# Subtypes and proliferation patterns of small intestine neuroendocrine tumors revealed by single-cell RNA sequencing

Einav Somech[1†], Debdatta Halder[1†], Avishay Spitzer[1†], Chaya Barbolin[1], Michael Tyler[1], Reut Halperin[2], Moshe Biton[3], Amit Tirosh[2*], Itay Tirosh[1*]

[1]Department of Molecular Cell Biology, Weizmann Institute of Science, Rehovot, Israel; [2]ENTIRE – Endocrine Neoplasia Translational Research Center, Division of Endocrinology, Diabetes and Metabolism, Chaim Sheba Medical Center, and Gray Faculty of Medical and Health Sciences, Tel Aviv University, Ramat Gan, Israel; [3]Department of Immunology and Regenerative Biology, Weizmann Institute of Science, Rehovot, Israel

**\*For correspondence:**
amit.tirosh@sheba.health.gov.il (AT);
itay.tirosh@weizmann.ac.il (IT)

[†]These authors contributed equally to this work

## eLife Assessment

This is a **valuable** study that uses single-cell RNA sequencing to define tumor-intrinsic transcriptional programs that characterize distinct types of small intestine neuroendocrine tumors. The evidence supporting the claims of the authors is **solid**, but would benefit from a larger sample size. The work will be of interest to cancer biologists studying neuroendocrine tumors, as well as those studying tumor heterogeneity more broadly.

**Abstract** Neuroendocrine tumors (NETs) occur primarily in the small intestine, lung, and pancreas. Due to their rarity compared to other malignancies in these organs, their complex biology remains poorly understood, including their oncogenesis, tumor composition, and the intriguing phenomena of mixed neuroendocrine non-neuroendocrine neoplasms (MiNEN). Here, we profiled ten low-grade small intestine NET (SiNET) samples as well as one mixed lung tumor by single-cell or single-nuclei RNA-seq. We find that SiNETs are largely separated into two distinct subtypes, in which the neuroendocrine cells upregulate epithelial or neuronal markers, respectively. Surprisingly, in both subtypes, the neuroendocrine cells are largely non-proliferative while higher proliferation is observed in multiple non-malignant cell types. Specifically, B and plasma cells are highly proliferative in the epithelial-like SiNET subtype, potentially reflecting the outcome of high Migration Inhibitory Factor (MIF) expression in those tumors, which may constitute a relevant target. Finally, our analysis of a mixed lung neuroendocrine tumor identifies a population of putative progenitor cells that may give rise to both neuroendocrine and non-neuroendocrine (squamous) cells, potentially explaining the origin of the mixed histology. Taken together, our results provide important insights and hypotheses regarding the biology of neuroendocrine neoplasms.

## Introduction

Neuroendocrine cells are present throughout the body and are thought to give rise to NETs in multiple organs. SiNETs have the second highest incidence, after lung NETs (*Dasari et al., 2017*). Histopathological grading of SiNETs is based on mitoses rate and on the Ki67 proliferation index, with low-grade comprising G1 and G2, and high-grade comprising G3. In addition, neuroendocrine carcinomas

(NECs) are included in the G3 category and have a poorly differentiated histopathological appearance (*Clift et al., 2020*).

The majority of SiNETs are low-grade well-differentiated tumors, usually diagnosed at advanced stages and often present with metastases and stage IV tumors, highlighting their high metastatic potential (*Niederle et al., 2010*). Interestingly, SiNETs are often multifocal, with a common finding of synchronous multiple primaries along the small intestine (*Gangi et al., 2018*; *Kalifi et al., 2021*). Two recent studies demonstrated that synchronous SiNET primaries within the same individual usually do not share somatic alterations, indicating that they arise from different cell clones (*Elias et al., 2021*; *Mäkinen et al., 2022*).

SiNETs are usually sporadic, with a surprisingly low mutational burden or recurrent mutations, and show mainly chromosomal alterations (*Hofving et al., 2018*; *Hofving et al., 2021*; *Scarpa, 2019*). The most frequent chromosomal aberration found in SiNETs is the loss of chromosome 18, as seen in 61–89% of patients (*Di Domenico et al., 2017*). In a cohort of 180 SiNETs, the most frequent mutation (found in 8% of tumors) was heterozygous frameshift mutations of *CDKN1B*, encoding the p27 cyclin regulator, suggesting a role of cell cycle deregulation in the progression of SiNETs (*Francis et al., 2013*). Further studies reported heterogeneity of the *CDKN1B* mutations even among lesions of the same patient (*Crona et al., 2015*). Overall, there is a limited frequency and consistency of genetic aberrations in SiNETs, of which mechanisms of initiation and progression remain poorly understood.

Another intriguing phenomenon with respect to the large group of neuroendocrine neoplasms (NENs) is that they are sometimes accompanied by a carcinoma component, resulting in 'mixed' neuroendocrine non-neuroendocrine neoplasms (MiNEN) (*Frizziero et al., 2020*). For a NEN to be termed mixed, it is necessary for each component to comprise at least 30% of the tumor, suggesting that many additional tumors may have both components but are not recognized as MiNEN. The origin of MiNENs remains unknown, and their complexity hinders conventional treatment approaches.

In this study, we sought to investigate the molecular characteristics of SiNETs to uncover novel insights into their oncogenic transformation, the resulting tumor ecosystems, and potential vulnerabilities. To this end, we applied single-cell transcriptomics to SiNETs as well as to one MiNEN.

## Results

### Single-cell and single-nuclei RNA-seq profiling of SiNETs

To understand the tumor ecosystem of low-grade SiNETs, we generated scRNA-seq profiles for ten primary tumor samples from eight patients using 10 x Chromium (*Supplementary file 1*). We profiled single cells from three fresh surgical samples (SiNET1-3) and single nuclei from the remaining seven frozen samples (SiNET4-10). After initial quality controls, we retained 29,198 cells from the ten samples (see Methods). For comparison, we also profiled one fresh small intestine adenocarcinoma (SiAdeno) and included it in our analyses.

We identified clusters of cells in each tumor and annotated them based on the expression of known marker genes as neuroendocrine (NE) cells, and as tumor microenvironment cells (TME) including T cells, B/plasma cells, macrophages, fibroblasts, endothelial cells, epithelial cells and natural killer (NK) cells (*Supplementary file 2*). While NK cells were detected only in one sample, the other six types of TME cells were each detected in multiple samples, and these exactly correspond to the TME cell types that are commonly detected in scRNA-seq analyses of other cancer types (*Gavish et al., 2023*). The clustering and marker expression of two exemplary SiNETs are shown in *Figure 1A–D*, and the remaining tumors are shown in *Figure 1—figure supplement 1*. Cell type proportions varied considerably between tumors (*Figure 1E*). In five tumors, NE cells were the most frequent, while other tumors had the highest frequency of other cell types, such as endothelial, fibroblast, epithelial, or T cells. The unique composition of each tumor sample likely reflects a combination of tumor-specific biology with spatial sampling within the tumor. Technical effects (e.g. single-cell analysis of fresh samples vs. frozen single nuclei analysis of frozen samples) could also impact the capture of distinct cell types, although we did not observe a clear pattern of such bias.

### Shared NE-specific genes define an SiNET transcriptomic signature

We next focused on the NE cells and defined both their common and their variable expression profiles across patients. To this end, we examined the eight samples (one per patient) in which we identified

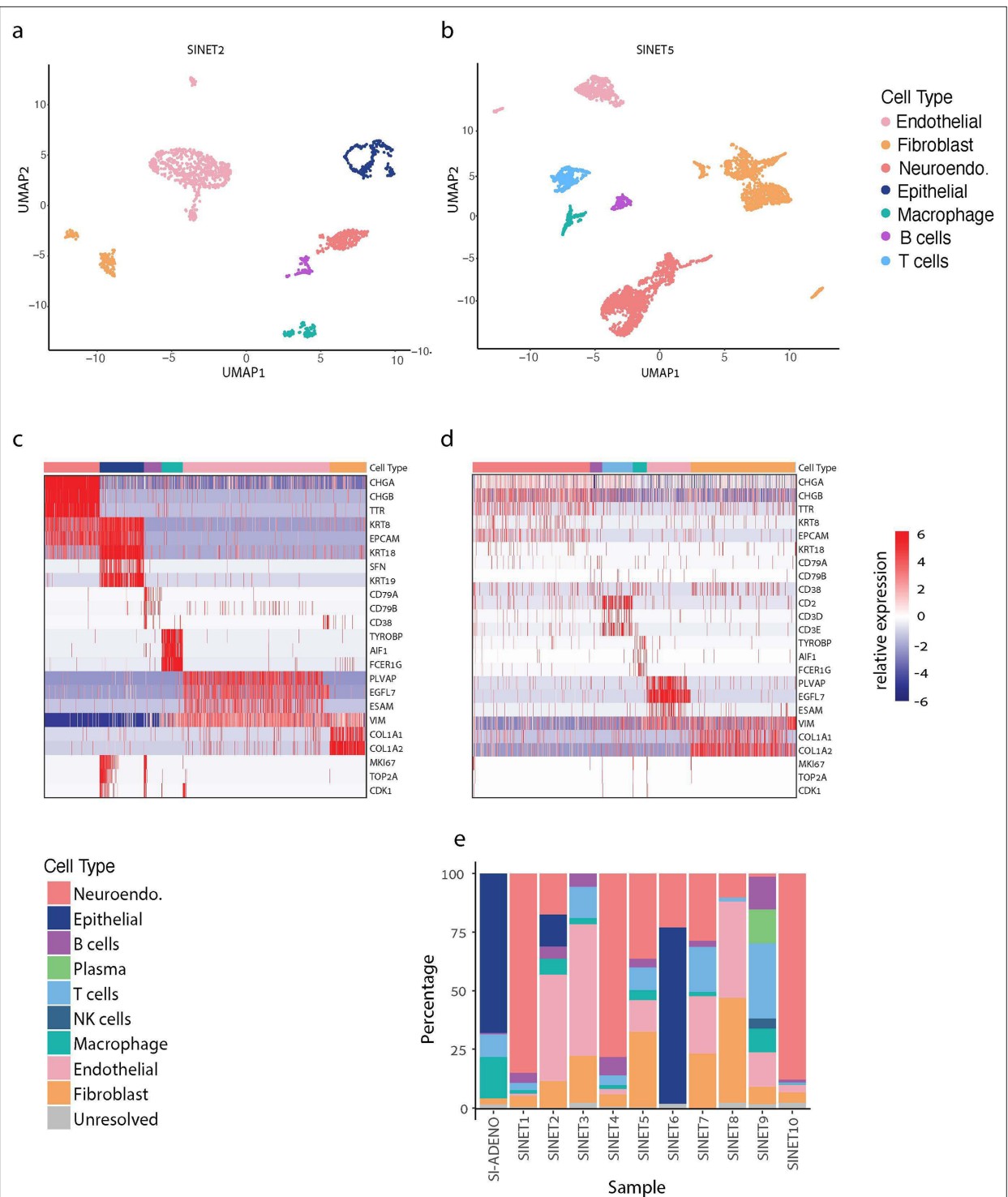

**Figure 1.** Cellular composition of small intestine neuroendocrine tumors (SiNETs) as determined by single-cell and single-nuclei sequencing. (**A, B**) UMAP plots showing the diversity of single cells from SiNET2 (**A**) and SiNET5 (**B**), colored by their cluster assignment. (**C, D**) Cluster annotations (top bar) in SiNET2 (**C**) and SiNET5 (**D**) are supported by the expression of canonical cell type markers (rows). Also shown are three cell cycle markers (bottom rows). (**E**) Cell type frequencies in each of the 10 SiNETs that we profiled, along with one SiAdeno sample.

The online version of this article includes the following figure supplement(s) for figure 1:

**Figure supplement 1.** Cellular composition of small intestine neuroendocrine tumors (SiNETs) as determined by single-cell and single-nuclei sequencing.

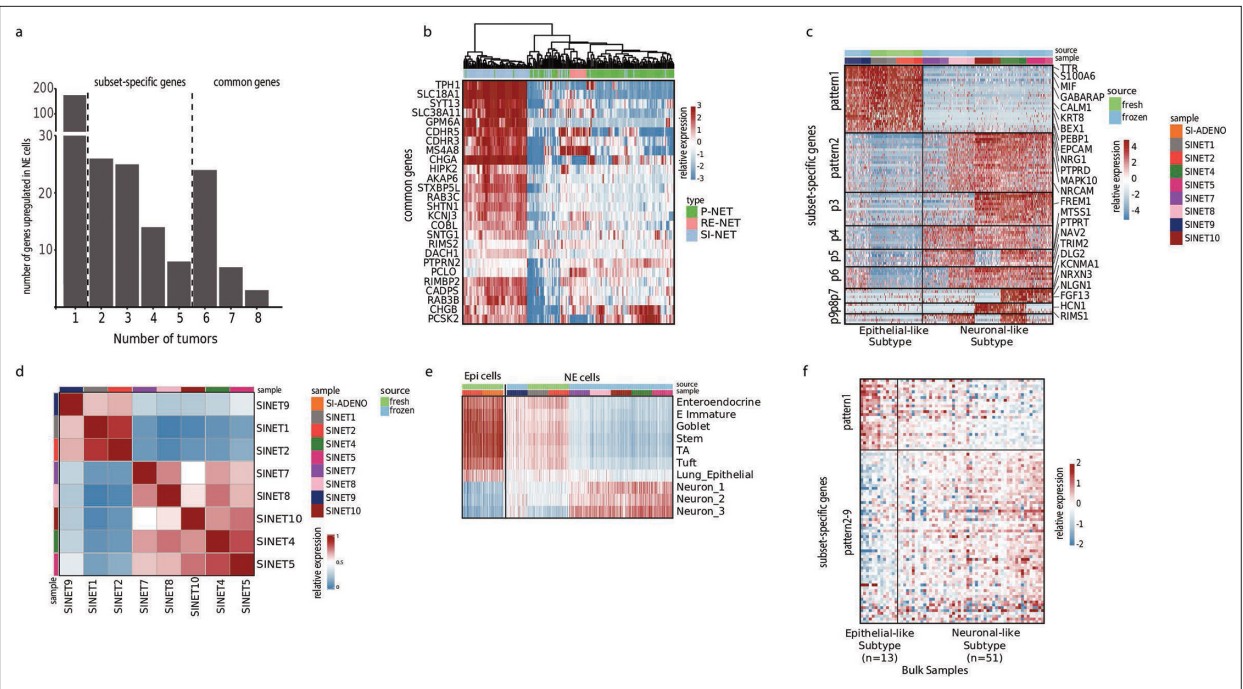

**Figure 2.** Small intestine neuroendocrine tumors (SiNETs) broadly classify into two major subtypes. (**A**) Bar plot showing the number of upregulated genes against a common threshold, number of genes (y-axis) vs number of tumors (x-axis). (**B**) Heat map showing a list of 25 representative genes that define SiNET signature of our single-nuclei RNA sequencing (scRNA-seq) cohort used to cluster neuroendocrine tumor (NET) samples from a bulk-seq dataset (*Alvarez et al., 2018*). Type of NET is color-coded on the top panel, with P-NET and RE-NET referring to pancreatic and rectal NETs. (**C**) Heatmap representing clustering of SiNET samples in our cohort based on genes that were differentially expressed and shared between 2–5 samples, showing two major variable gene programs. (**D**) Correlation heat map between the NET samples. (**E**) Heatmap showing average expression of epithelial and neuronal gene sets (rows) in the neuroendocrine and epithelial cells from our SiNET samples (columns). Epithelial gene sets include signatures of multiple cell types from the small intestine (*Haber et al., 2017*), and neuronal gene sets include three clusters of neurons (*Mahalingam et al., 2020*). (**F**) Heatmap of the SiNET-dominated cluster from the bulk dataset (*Alvarez et al., 2018*) was subjected to differential expression analysis using the same set of genes as (**C**).

The online version of this article includes the following figure supplement(s) for figure 2:

**Figure supplement 1.** Specific upregulated genes expressed in neuroendocrine cells per small intestine neuroendocrine tumor (SiNET) sample.

NE cells. We first compared NE cells from each tumor to non-NE cells from the same tumor and identified all genes that are highly upregulated in NE cells. Twenty-six genes were consistently upregulated in NE cells from most tumors (n>5) and hence were defined as common NE-specific genes (*Figure 2A*).

The common NE-specific genes could potentially serve as a transcriptomic signature for SiNETs. However, they may also include generic markers of NETs that are not unique to SiNETs. To examine the specificity of this signature, we examined their relative expression in a bulk RNA-seq dataset that contains NETs of the small intestine, pancreas, and rectum (*Alvarez et al., 2018*). Almost all signature genes were preferentially upregulated in NETs of the small intestine compared to the other NETs, suggesting that they reflect an efficient SiNET signature (*Figure 2B*, *Supplementary file 3*). The signature genes include known markers of SiNET (*Andersson et al., 2016*) as well as of enterochromaffin cells (*Haber et al., 2017*), such as *CHGA*, *CHGB*, and *TPH1*. While some of these markers are not unique to SiNETs, the bulk RNA-seq data suggests that, at least at the mRNA level, they are higher expressed in SiNETs than in other NETs. These genes were most significantly enriched with gene sets associated with neuroendocrine-related functions such as exocytosis, neurosecretory vesicle, and serotonergic synapse (all with adjusted p<0.001).

## Heterogeneity of NE cells reveals two SiNET subtypes

However, many more genes were found as upregulated in NE cells of only a few of the SiNETs, and these were separated into tumor-specific (n=1, see *Figure 2—figure supplement 1*) and subset-specific

(between 2 and 5 tumors). Subset-specific genes may help to uncover functionally distinct subtypes of SiNETs and, therefore, we investigated them further. These genes exhibited nine distinct expression patterns across eight SiNET samples from eight distinct patients (*Figure 2C*, *Supplementary file 3*). Notably, pattern #1, which was associated with the largest number of genes (n=26), was highly expressed in three tumors that had low expression of all other eight patterns. In contrast, the other five tumors highly expressed genes from multiple patterns and had low expression of pattern #1 genes. This analysis highlighted a division of the eight SiNETs into two primary subtypes based on pattern #1 vs. all other patterns. Notably, the same subtype division was clearly detected by clustering of the SiNETs based on the global expression profiles of NE cells in each tumor (*Figure 2D*).

The first subtype (SiNETs 1, 2, and 9) was associated with upregulation of 129 genes, including the epithelial markers *EPCAM* and *KRT8* (*Supplementary file 4*). While this subtype included the two fresh samples (SiNETs 1 and 2), it also included one frozen sample (SiNET9) and, therefore, is unlikely to reflect technical confounders. The second subtype (SiNETs 4, 5, 7, 8, and 10) was associated with upregulation of 73 genes, including neuronal-related genes such as *NAV3, RYR2, CACNA1A/B*, and *KCNIP1*. This led us to speculate that even though both subtypes express the neuroendocrine markers described above (i.e. common genes), the first subtype may have higher overall similarity to epithelial cells compared to the second subtype, which might have more neuronal features.

To examine this possibility, we evaluated the expression of other cell type signatures across NE cells from the two SiNET subtypes (*Figure 2E*). Signatures of multiple epithelial cell types from the small intestine or from the lung, including those of enteroendocrine cells, were expressed more highly in subtype 1 than in subtype 2 SiNETs. Nevertheless, the expression of such epithelial signatures was considerably lower in NE cells from subtype 1 tumors than in bona fide epithelial cells such as the non-malignant epithelial cells identified in SiNET6 or the malignant cells in the SiAdeno tumor. Thus, epithelial signatures are upregulated in subtype 1 tumors but not at the same level as in bona fide epithelial cells. Conversely, signatures of neuronal cell types had the opposite expression pattern – highest in subtype 2 SiNETs, intermediate in subtype 1 SiNETs, and lowest in bona fide epithelial cells. Based on these results, we renamed subtypes 1 and 2 as the *epithelial-like* and *neuronal-like* SiNET subtypes.

To validate the existence of these two SiNET subtypes, we turned to analyze an external bulk RNA-seq dataset of SiNETs (*Alvarez et al., 2018*). The signal of NE cells is diluted in bulk RNA-seq due to the profiling of entire tumor samples, and especially given the wide diversity of cell type compositions that we observe in single-cell analysis (see *Figure 1E*). However, we wondered whether we would still be able to detect the two subtypes when directly analyzing the subset-specific genes defined above. Indeed, the 81 SiNET bulk profiles could be separated into the *epithelial-like* and *neuronal-like* subtypes based on expression of pattern #1 genes vs. the genes of the other patterns (*Figure 2F*). Moreover, the proportion of the two subtypes (20% vs 80%) was comparable to those seen in our single-cell cohort (30% vs 70%). This analysis supports the generality of the two SiNET subtypes and their independence from the potential confounding effects of fresh vs. frozen samples in our data, which do not affect the bulk RNA-seq dataset.

## Heterogeneity in the SiNET tumor microenvironment

Next, we examined the diversity of cellular states within each non-NE cell type of the tumor micro-environment (TME). For each cell type, we analyzed the diversity within each tumor, searching for distinguishable subpopulations of cells, their occurrence across tumors, and their potential functional implications (see *Figure 3*). Below, we briefly describe the three most notable cases of diversity within cell types that we observed.

Among fibroblasts, we found the highest diversity in SiNET8, with three distinguishable subpopulations (*Figure 3A*). To understand their differences, we both conducted a differential expression analysis (*Figure 3B*) and compared their profiles to recently described fibroblast signatures (*Gavish et al., 2023*; *Figure 3C*). This analysis indicated that cluster 1 resembles other cancer-associated fibroblasts (CAFs), while clusters 2 and 3 are distinguished by specific expression profiles. Cluster 3 cells were primarily distinguished by expression of MHC-II genes and are thus consistent with previous observations of antigen-presenting CAFs (*Elyada et al., 2019*). Cluster 2 cells upregulate several programs of pericytes and myofibroblasts but are further distinguished by upregulation of four ABC-transporters (ABCA6, ABCA8, ABCA9, and ABCA10). To our knowledge, such consistent upregulation of multiple

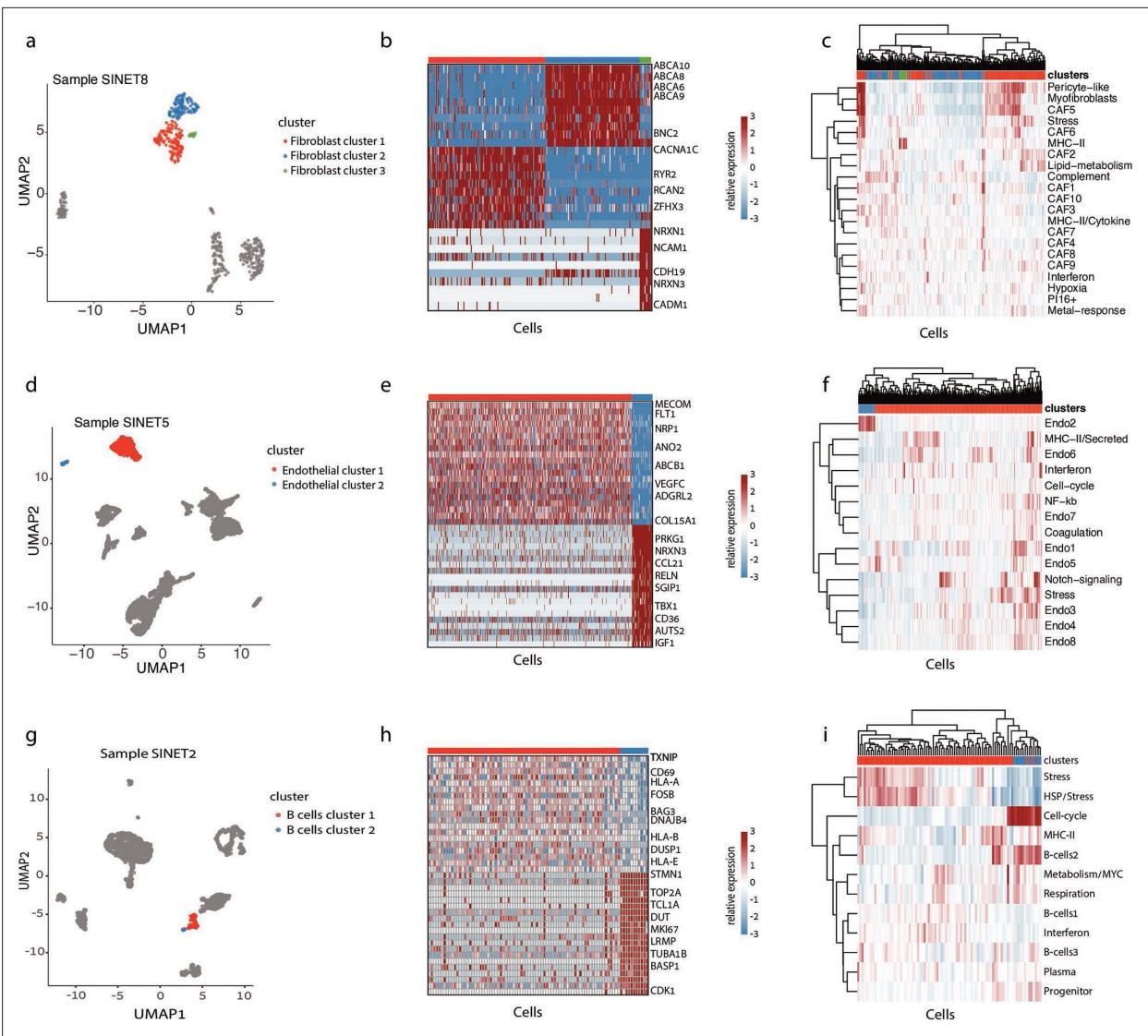

**Figure 3.** Heterogeneity in the small intestine neuroendocrine tumor (SiNET) microenvironment. For each of the three non-malignant cell types, the diversity of that cell type is shown in one exemplary tumor: fibroblast heterogeneity in SiNET8 is shown in (**A–C**), endothelial cell heterogeneity in SiNET5 is shown in (**D–F**), and B-cell heterogeneity in SiNET2 is shown in (**G–I**). For each cell type, three panels depict three types of analyses. The first panel (**A, D, G**) is a UMAP plot of the respective tumor, where only the respective cell type is colored, and distinct colors highlight the clusters of that cell type. The second panel (**B, E, H**) shows differential expression analysis between the first two clusters using heatmaps, with labeling of selected genes. The third panel (**C, F, I**) shows clustering of cells from that cell type (columns) based on their relative expression of previously defined (**Gavish et al., 2023**) signatures of diversity in that cell type (rows); the top panel shows assignment of cells to clusters.

The online version of this article includes the following figure supplement(s) for figure 3:

**Figure supplement 1.** Heterogeneity in the small intestine neuroendocrine tumor (SiNET) microenvironment.

ABC transporters was not observed in previous analysis of CAF heterogeneity (**Lavie et al., 2022**; **Luo et al., 2022**). Yet, in our data, upregulation of ABC transporters was also detected in a subset of CAFs from SiNET3 and SiNET5 (**Figure 3—figure supplement 1A–D**). These results suggest a unique feature of CAFs in a subset of SiNETs, possibly due to the unique microenvironment of SiNETs.

Among endothelial cells, we found two highly distinct subpopulations in SiNET5 (**Figure 3D–E**). Comparison to pan-cancer endothelial signatures mapped the smaller subpopulation to the Endo2 poorly described signature that was detected primarily in lung and skin tumors (**Gavish et al., 2023**; **Figure 3F**). Similar subpopulations of endothelial cells were also detected in SiNET2, highlighting the Endo2 signature as a recurrent pattern of subsets of SiNET endothelial cells (**Figure 3—figure supplement 1E–F**).

Among B cells, two distinct populations were found in SiNET2 (*Figure 3G*). Differential expression showed that the small B cell subpopulation is distinguished by upregulation of many cell cycle genes, including canonical markers (MKI67, TOP2A, CDK1), reflecting proliferating B cells (*Figure 3H–I*). Moreover, we noticed that, among all SiNET2 cells, the canonical cell cycle markers are expressed in B cells more than in all other cell types, including the malignant NE cells (*Figure 1C*). This observation prompted us to turn our attention to cell cycle patterns across all cell types within the SiNETs.

## Proliferation of NE and immune cells in SiNETs

Most SiNETs are low-grade tumors with a low mitotic index, but the exact identity of proliferating cells in SiNETs is unknown. We used previously defined signatures of the cell cycle to identify all cycling cells. Notably, as we and others demonstrated previously (*Gavish et al., 2023*; *Puram et al., 2023*;

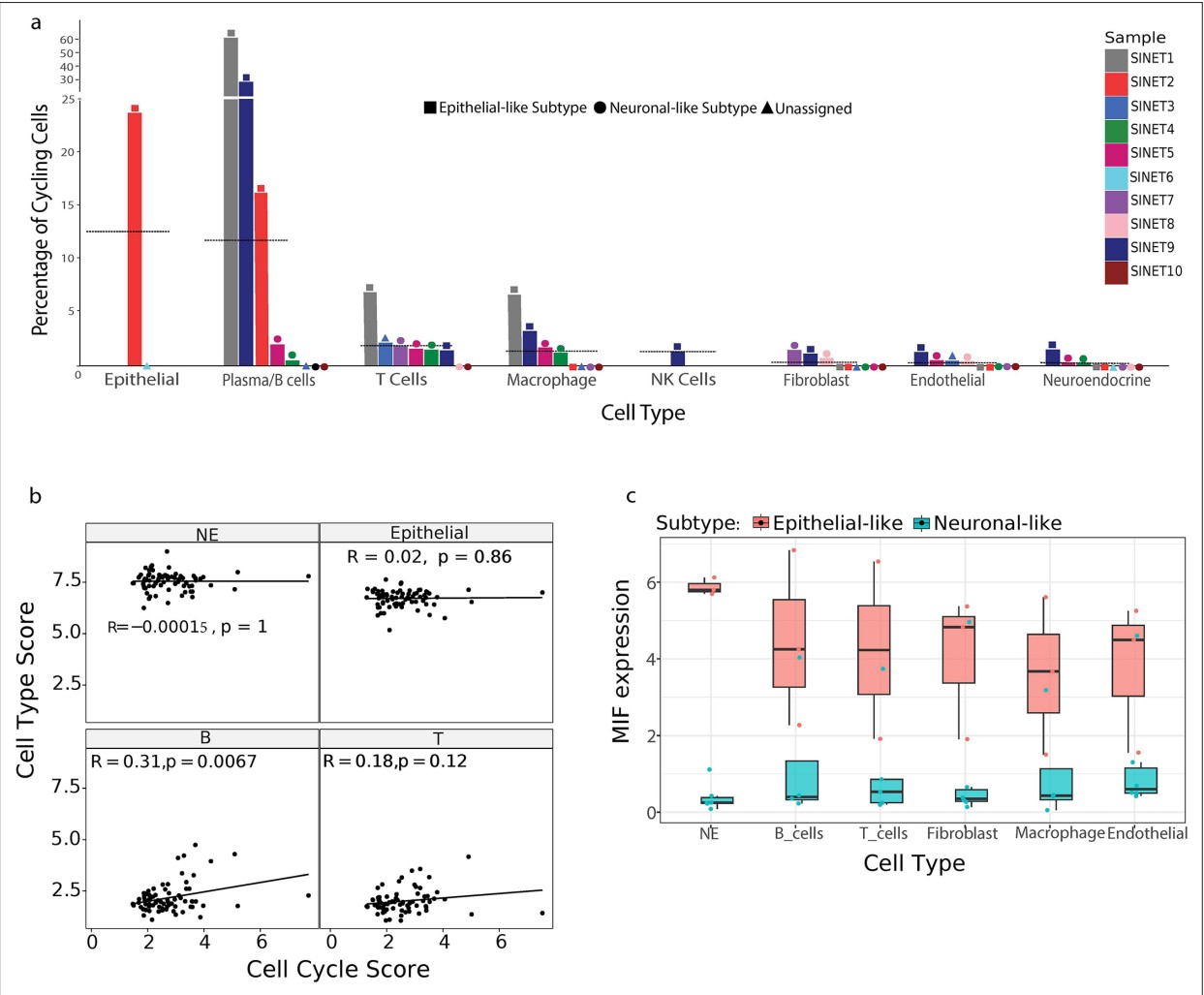

**Figure 4.** Cell cycle analysis reveals proliferating B cells in small intestine neuroendocrine tumors (SiNETs). (**A**) Bars show the percentage of cycling cells (y-axis) per cell type and per tumor (x-axis). Tumors are color-coded, the two subtypes, epithelial-like and neuronal-like, are differentiated by distinct shapes, represented as square and circle, respectively. Information regarding the presence of cell types with zero percentage of cycling cells is provided along the x-axis. Horizontal lines indicate average percentages of cycling cells per cell type. (**B**) Correlation between cell type and cell-cycle program as computed from an SiNET bulk RNA-seq dataset (*Alvarez et al., 2018*). Score for each cell type is represented at the top of individual panels. (**C**) Boxplot depicting the expression of migration inhibitory factor (MIF) in each SiNET cell type, for each of the two SiNET subtypes.

The online version of this article includes the following figure supplement(s) for figure 4:

**Figure supplement 1.** Heat map illustrating the expression of G1/S and G2/M genes across various cell types in the Epithelial-like Subtype (**A**), Neuronal-like Subtype (**B**) and the siAdeno sample (**C**).

**Figure supplement 2.** Scatter plot illustrates the percentage of cycling B/Plasma cells and the correlation between the germinal center signature and cycling B/Plasma cells signature.

*Kowalczyk et al., 2015*), cell cycle involves the consistent upregulation of dozens of canonical genes and, therefore, the cycling cells can be robustly detected by scRNA-seq along with their phase along the cell cycle (*Figure 4—figure supplement 1*).

Surprisingly, extremely few cycling cells were observed among the malignant NE cells (0.246% on average) (*Figure 4A*). This fraction is considerably lower than what we detect in other cancer types (*Puram et al., 2023*), and as an extreme example from the same dataset, in the SiAdeno sample, we find ~13% of cycling epithelial cells using the same method (*Figure 4—figure supplement 1*). Notably, the fraction of SiNET cycling cells was lower for NE cells than for all other cell types identified in the SiNET ecosystem (*Figure 4A*; see also *Figure 1C and D*). Relatively high fractions of cycling cells (>10%) were found only in epithelial or in B/plasma cells. Epithelial cells were only detected in two tumors, and of those only in one tumor they have a high fraction of cycling cells. In contrast, B/plasma cells were detected more commonly – in eight tumors – and in three of those, they had a high fraction of cycling cells. The other cell types all had low-to-intermediate fractions of cycling cells across all SiNETs, but even those fractions were consistently higher than for the NE cells.

Thus, at least in our SiNET cohort, proliferation is associated with the tumor microenvironment rather than with the NE cells. Notably, both the low proliferation of NE cells and the much higher proliferation of immune cells (especially B/plasma cells, but also T cells and macrophages) were consistently observed in both fresh samples analyzed by single-cell RNA-seq (SiNET1 and 2) and frozen samples analyzed by single nuclei RNA-seq (SiNET9) and, therefore, are independent of platform and the potential biases in capture of specific cell types. This result raises intriguing questions regarding the manner by which SiNETs grow, and the meaning of their mitotic index (see Discussion).

Next, to ensure that this result is not unique to our cohort, we reanalyzed bulk RNA-seq data of 81 SiNET samples (*Alvarez et al., 2018*). We reasoned that if NE cells are a major source of proliferation, then cell cycle signatures should correlate with the expression of NE marker genes across bulk SiNET samples. Similarly, if other cell types, such as B cells or T cells, are more proliferative than the NE cells, then their markers should correlate with the cell cycle signature. As expected from the single-cell analysis, we found that the cell cycle signature significantly correlates positively with B cell markers, and to a more limited degree with T cell markers, but not with NE markers (*Figure 4B*), supporting the broader relevance of our results.

## B cell proliferation and MIF upregulation in the epithelial-like SiNET subtype

One potential explanation for the high proliferation of B cells is that our SiNET samples may have included germinal centers (GC), in which B cells are expected to proliferate. To assess this possibility, we scored the B cells for a previously defined GC signature (*Brescia et al., 2018*). While two tumors with high B cell proliferation also had high GC scores, this was not the case for the third tumor (*Figure 4—figure supplement 2*). Thus, the inclusion of GCs may partially explain the unusual proliferation of B cells.

An alternative possibility is that B cells proliferate in response to particular signals in the SiNET microenvironment, for example, due to factors secreted by other cells. The proliferation of B/plasma cells was high only in three of the SiNETs and was absent or low in five other SiNETs (*Figure 4A*). Intriguingly, all of the epithelial-like SiNETs had high B/plasma cell proliferation, and all of the neuronal-like SiNETs (in which B/plasma cells were detected) had low proliferation of those cells. This perfect agreement with SiNET subtypes suggests that the unique features of the epithelial-like subtype may drive the proliferation of B/plasma cells. To identify such features, we defined the differential expression between the two subtypes for each cell type (*Supplementary file 4*).

This analysis highlighted Macrophage Migration Inhibitory Factor (MIF) as a prominent marker of the epithelial-like subtype that could potentially drive the proliferation of B/plasma cells. First, MIF was one of the top markers of the epithelial-like subtype in our analysis of NE cells (*Figure 2C*), and a closer inspection reveals an extreme degree of differential expression, with almost no detected reads in the neuronal-like subtype and a consistently high expression in the epithelial-like subtype (*Figure 4C*). Second, upregulation of MIF in the epithelial-like subtype was also detected in all other cell types, although the effect was strongest in NE cells (*Figure 4C*). Such consistent upregulation across six different cell types is not seen for any other genes, highlighting the unique upregulation of MIF in the ecosystem of the epithelial SiNETs and supporting its potential causal effect in driving other

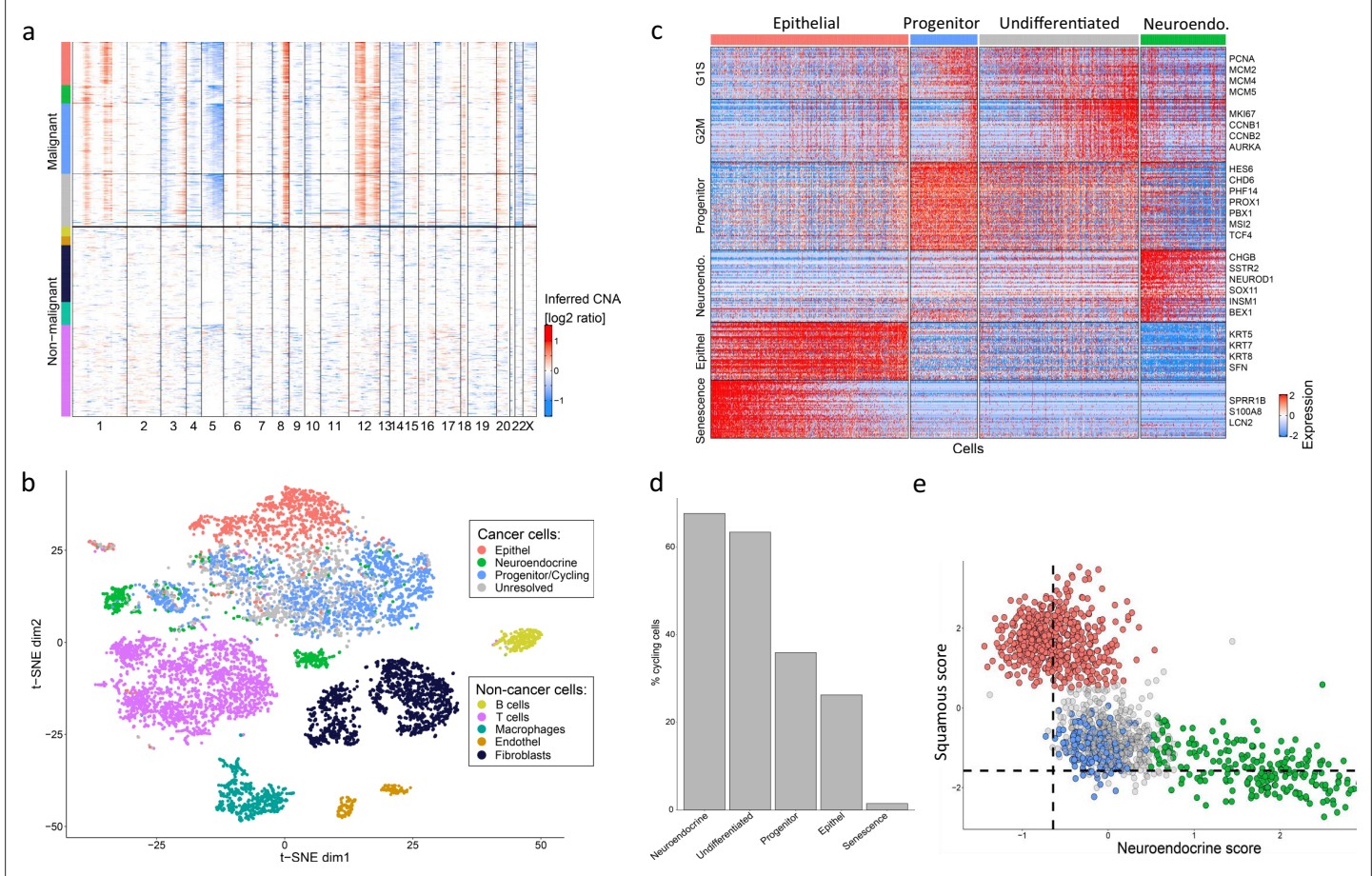

**Figure 5.** A putative progenitor population in mixed Large Cell Neuroendocrine Carcinoma (LCNEC). (**A**) Copy number variation (CNV) profiles inferred from single-nuclei RNA sequencing (scRNA-seq) data for all cells from the LCNEC sample. Malignant and non-malignant cells are annotated based on their CNV profiles, using the same color codes as in the next panels. (**B**) tSNE plot showing the diversity of single cells from the mixed lung tumor, colored by their clustering. (**C**) Heatmap shows relative expression of differentially expressed genes (rows), separated by horizontal lines into programs that distinguish between the four populations of cells detected in the LCNEC sample. Also included are two cell cycle programs (G1/S and G2/M). Columns correspond to malignant cells, separated into the four populations by vertical lines and as indicated by color at the top. Selected genes are labeled for each program. (**D**) Bars show the percentage of cycling cells in each malignant cluster. (**E**) Malignant cells scored against an epithelial vs. neuroendocrine program (gene set), colored by their assignment into four populations.

phenotypes such as B/plasma cell proliferation. Third, MIF is an important cytokine that binds CD74 and was previously shown to regulate B cell proliferation by multiple studies (*Gore et al., 2008*; *Noe and Mitchell, 2020*).

## Putative progenitors in mixed tumors

Apart from our focus on SiNETs, we were intrigued by the phenomena of MiNEN and aimed to profile such tumors by single-cell RNA-seq. Due to their rarity and the difficulty of recognizing mixed tumors prior to their surgery, we were so far able to profile only one such tumor. This tumor was a Large Cell Neuroendocrine Carcinoma of the lung (LCNEC), mixed with a squamous cell carcinoma. While this tumor is highly different from SiNETs, we believe that this N-of-1 case study is of interest and raises important questions that might also be of relevance for SiNET and other NENs, and hence we describe its analysis below.

We profiled the fresh mixed-LCNEC tumor sample by scRNA-seq using the 10x Chromium. Cells were first classified as malignant or non-malignant, based on inference of copy-number aberrations, as described previously (*Tirosh et al., 2016*; *Figure 5A*). Clustering of the non-malignant cells and annotation of the clusters based on standard markers separated them into fibroblasts, endothelial cells, macrophages, T cells, and B cells (*Figure 5B*).

We found extensive diversity among the malignant cells, which highlighted several subpopulations of cells (*Figure 5B–C*). These include the two malignant components expected based on the classification as a mixed tumor – neuroendocrine cells (e.g. expressing CHGB and SSTR2) and epithelial squamous cells (e.g. expressing KRT5 and KRT8). The epithelial cells further varied in their expression of a program that we previously termed 'epithelial senescence' (*Kinker et al., 2020*), which was indeed associated with lack of proliferation (*Figure 5D*). Interestingly, we also found additional subpopulations of malignant cells with intermediate expression levels of both the neuroendocrine genes and the squamous genes (*Figure 5E*). These 'intermediate' cells included two clusters that we termed *undifferentiated* and *progenitors*, since they were distinguished by expression of developmental and stemness-related regulators (*Figure 5C*). These include the homeobox transcription factors PBX1 and PROX1 (*Chan et al., 2009*; *Muggeo et al., 2021*), the RNA-binding protein MSI2 (*Hope and Sauvageau, 2011*), the WNT-related TCF4 (*Lu et al., 2012*; *Sun et al., 2020*), and the NOTCH-related transcription factor HES6 (*Krossa et al., 2022*; *Supplementary file 5*). All four malignant populations were highly proliferative, except for the subset of squamous cells that expressed the epithelial senescence program (*Figure 5D*). The progenitor cluster, with intermediate expression of the two lineage programs and upregulation of stemness factors, is consistent with the possibility that stem/progenitor cells may give rise to the two differentiated components that constitute the mixed phenotype. However, additional MiNENs would need to be profiled at single-cell resolution to explore the generality of this observation.

## Discussion

scRNA-seq is a promising approach to improve our understanding of SiNET biology, and in particular, their composition and diversity. A recent study described scRNA-seq analysis of NETs but only had samples from two SiNET patients and had enrichment of immune cells with a limited amount of NE cells (*Hoffman et al., 2023*). Here, we profiled at single-cell resolution ten SiNET samples from eight patients and focused our analysis primarily on the NE cells. While low-grade SiNETs are known to have a low mitotic index, any cancer is driven by cell proliferation such that malignant cells are expected to have higher cell cycle activity than non-malignant cells. Surprisingly, this does not seem to be the case in our analysis of SiNETs, where malignant cells appear to be less proliferative than multiple types of non-malignant cells. In such cases, it is conceivable that pathological evaluation of tumor proliferation (Ki67 counts) may be confounded by non-malignant cycling cells and, therefore, inflated. But a more fundamental question is how could those tumors initiate and progress when the NE cells appear to be largely non-proliferative?

We can envision multiple potential answers, although further studies are needed to resolve these possibilities. First, since growth is determined by the balance between cellular proliferation, death, and migration, SiNETs may still grow despite their very low proliferation through abnormally low death or abnormal patterns of migration. We are not aware of any evidence in support of this possibility, but we cannot exclude it. Second, the proliferation of NE cells may be inhibited by prior treatments with somatostatin analogues. Third, proliferation may be spatially or temporally restricted. Accordingly, higher proliferation may occur only in particular niches or at a particular time of tumor progression. This possibility could be examined in future work using spatial omics.

A related possibility is that proliferation may be restricted to a hidden stem/progenitor population that we have not captured, either because it is rare or because it may be sensitive to our experimental workflows. Such proliferative stem/progenitor cells were described in other cancer types and are consistent with the cancer stem cell hypothesis. For example, we previously described the cellular hierarchy of oligodendroglioma, in which proliferation is restricted to cells resembling neural progenitors, while the tumor is dominated by more differentiated cells resembling oligodendrocytes and astrocytes (*Tirosh et al., 2016*). A second, and more relevant example, is presented here with the N-of-1 analysis of a mixed LCNEC. In this case, high proliferation is seen across all malignant components, but the mere presence of a proliferative stem/progenitor population suggests that bipotent progenitors may explain the diversity within mixed LCNECs and possibly even the wider phenomena of MiNENs. The existence and the properties of such bipotent progenitors in MiNENs would require further validation by future studies, yet it is tempting to speculate that such progenitors may also exist in other neuroendocrine tumors (e.g. SiNETs).

SiNETs deviate from classical oncogenesis not only by their low malignant cell proliferation, but also by their paucity of mutations and apparent driver events. Accordingly, previous work proposed that SiNETs may be driven by epigenetic aberrations in NE cells or by effects of the TME. Our observation of minimal proliferation in NE cells supports the latter possibility of TME oncogenic effects. In particular, SiNETs of the epithelial-like subtype are associated with extremely high levels of MIF, an important and pleiotropic cytokine that may influence the TME in multiple important ways and possibly create a unique inflammatory TME. One potential effect that we observe is high proliferation of B or plasma cells. Notably, the three SiNETs of the epithelial-like subtype are not associated with higher numbers of B/plasma cells, compared to other SiNETs, but rather only with increased cell cycle of the B/plasma cells, suggesting that their proliferation is balanced by high cell death. The implications of high B/plasma cell turnover and of other downstream effects of high MIF expression are unclear, but raise the possibility that MIF-CD74 interaction may constitute a relevant target for the epithelial-like SiNET subtype. Notably, MIF-CD74 inhibitors have already been developed and initially tested for the treatment of solid tumors (*Mahalingam et al., 2020*; *Varinelli et al., 2015*; *O'Reilly et al., 2016*).

In summary, single-cell profiling of SiNETs revealed two distinct subtypes that differ in NE profiles and in the proliferation of B-cells, possibly justifying different therapeutic strategies (e.g. MIF inhibition). The two subtypes share a minimal proliferation of NE cells, which raises questions about the initiation and growth of SiNETs and the potential role of TME cells, in line with the limited mutations observed in SiNETs. Our analysis of a mixed tumor further suggests the potential presence of bipotent progenitors that may account for mixed phenotypes. Taken together, these results improve our understanding of neuroendocrine neoplasms while highlighting their complex biology.

## Methods

### Human samples

The study was approved by the Institutional Review Board of Sheba Medical Center (Ethical Approval Identifier: SMC-18–5674). Informed consent was obtained from each patient prior to any study-related procedures. Patients received a comprehensive explanation of the study's objectives and methods, were given ample time to consider their participation, and had the opportunity to ask questions and receive thorough answers before signing the consent form. Following the consent process, all samples and data were fully anonymized. Each participant was assigned a unique code, which was used throughout the study and was kept separate from any identifying information. As a result, the dataset is fully anonymized, and its use and publication do not require additional consent.

For tissue preparation for immunohistochemistry, flash-frozen SiNETs were stored at −80 °C until cutting. Blocks were prepared in OCT followed by freezing and sectioning. Frozen tissue was sectioned using a −20 °C temperature on a cryostat (Leica CM3050 S) onto microscopic slides (Thermo, Superfrost Plus).

### Tumor dissociation and library preparation

Fresh tumors were collected directly from the operating room at the time of surgery. Tumors were washed in ice-cold HBSS, minced using a pair of Noyes spring scissors (Fine Science Tools) and then enzymatically dissociated using a tumor dissociation kit (Miltenyi Biotec) in a GentleMACS Octodissociator on low speed settings (*Tirosh et al., 2016*). Single-cell suspension was obtained by passing dissociated slurry through a 60–100 micron cell strainer (Miltenyi) and the filtrate was subjected to RBC removal using a RBC lysis buffer (Roche). Dead cell removal (Miltenyi) was performed on cell suspension following the manufacturer's protocol, and cells were assessed for viability and density. 5 µl of Trypan blue (Thermo Fisher Scientific) was mixed with 5 µl of the sample and loaded onto a chip disposable automated hemocytometer (Countess II). Cell concentration was adjusted such that a total of 8000–10,000 cells were loaded onto each channel of the 10 x Genomics Single-Cell Chromium Controller.

For frozen tumors, a similar dissociation procedure was performed with the following modifications. Nuclei isolation was achieved using either ST-based buffers as done before (Slyper et al.) or by EZ-lysis method (Sigma). Briefly, tissue samples were cut into pieces <0.5 cm using spring Noyes scissors and then homogenized in ice-cold buffer (ST-based or EZ lysis) using a glass Dounce tissue grinder (Sigma). Nuclei were centrifuged at 500 g, with low acceleration/deceleration for 5 min at

4 °C, washed with 5 ml ice-cold ST/EZ lysis buffer, and incubated on ice for 5 min. A suspension buffer containing PBS, BSA, and RNAse inhibitor was used to wash and resuspend the nuclei pellet. Nuclei suspension was filtered through a 40 µm cell strainer (Miltenyi) and counted. A final concentration of 1000 nuclei per µl was used for loading on a 10 x channel with a target of 10,000 nuclei per channel in the 10 x controller.

The Chromium Next GEM Single-Cell 3′ GEM, Library & Gel Bead Kit v3.1, Chromium Single-Cell 3′ Feature Barcode Library Kit, Chromium Next GEM Chip G, and 10 x Chromium Controller (10 x Genomics) was used for generation of libraries of fresh tumors/single cells and the Chromium Next GEM Single 5′ GEM was used for frozen tumors/single-nuclei.

As per the standardized protocol of creating 10 x libraries, single cells, reagents, and single gel beads containing barcoded oligonucleotides were emulsified and encapsulated into nanoliter-sized droplets followed by reverse transcription (RT). Following the manufacturer's recommendations, RT samples were subjected to cDNA amplification, fragmentation, adapter, and sample index attachment. Libraries from two 10 x channels were pooled together and sequenced on one lane of an Illumina NovaSeq-6000, using an sp-100 kit or 4 were pooled together on two lanes with an S1-100 kit, with paired end reads as follows: read 1, 26 nt; read 2, 55 nt; index 1, 8 nt; index 2, 0 nt.

## Sample normalization, filtering, and annotations

Four samples (SiAdeno, SiNETs 1–3) underwent single nuclei sequencing, while the remaining samples (SiNETs 4–10 and the LCNEC sample) were subjected to scRNA-seq. To facilitate comparative analysis, we converted counts of unique molecular identifier (UMI) counts to transcripts per million (TPM) values, which were divided by 10 since the actual complexity of cells is assumed to be in the realm of ~100,000 transcripts and not 1 million as implied by the TPM measures. The resulting values were added to 1 and log2-transformed. Finally, to derive relative expression, we centered the value of each gene in each tumor, except in the analysis, where we were interested in absolute expression levels.

To ensure data quality, we excluded low-quality cells based on the number of detected genes. For the nuclei-seq platform, we used a threshold of 1000 detected genes, while for the scRNA-seq platform, we used a threshold of 700 in order to retain lymphocytes that tend to have low numbers of detected genes. Additionally, we performed most analyses with a filtered set of genes, retaining only the genes in which the logged row-means per gene across all cells passed 4.5. After filtering, the number of cells analyzed per sample ranged from 287 to 10,802.

For further downstream analyses, we have avoided the use of integration methods as we believe that they tend to distort the data and decrease tumor-specific signals. Instead, we primarily analyzed one tumor at a time and never directly compared cell profiles across distinct tumors but only compared the differences between subpopulations in a given tumor; specifically, as described below, we normalized the expression of NE cells by subtracting the expression of reference non-NE cells from the same tumor as a method to decrease batch effects.

## Neuroendocrine cell signatures and their classification

To identify NE-specific genes, we performed a differential expression analysis between NE cells and a reference group consisting of Macrophages, Fibroblasts, and Endothelial cells. For each sample, we sampled 50 NE cells along with 50 cells from the reference group. Differentially expressed (DE) genes were defined as those with a fold change greater than 8 and a p-value lower than 0.05. False discovery rate (FDR) correction was applied to the p-values of individual genes to account for multiple comparisons. In samples that had insufficient reference cells (SiNET7 and SiNET8), we sampled the reference cells from a different sample that exhibited the highest correlation with the given sample. Samples that did not contain NE cells (SiNET3 and SiAdeno) or had too few NE cells (SiNET6), based on a defined threshold of 100 cells, were excluded from this analysis. Accordingly, SiNET9 was included in the analysis due to its high absolute number of profiled cells (130), which resulted in a sufficient count of NE cells, despite having a relatively small fraction of NE cells.

We classified the DE genes into three groups: Common genes, Subset-specific genes, and Sample-specific genes. Common genes were defined as those differentially expressed in at least six samples, Subset-specific genes were those differentially expressed in 2–5 samples, and Sample-specific genes were differentially expressed only in one sample.

## Clustering of subset-specific NE genes into nine patterns

We divided the subset-specific NE genes based on their expression pattern across the eight tumors that contained enough NE cells. To this end, we first considered all potential binary patterns, in which a gene is considered as either expressed (1) or not expressed (0) in each tumor. This defines $2^8=256$ theoretical patterns. Next, for each gene, we calculated the correlation between its relative expression across the eight samples, and each of the 256 binary patterns; the gene was then assigned to the pattern with the highest correlation. This uncovered nine patterns that were each assigned to >5 genes (*Figure 2C*,*Supplementary file 3*).

## Cell cycle analysis

To investigate the cell cycle dynamics within the dataset, we utilized a canonical cell cycle gene set (*Tirosh et al., 2016*). Each cell in every sample was scored based on the expression of these cell cycle genes. To identify cycling cells, we established a threshold based on the distribution of scores within each sample. We chose a relatively lenient threshold (log2 fold change of 1.5), and further verified that our main claims are qualitatively maintained when using a more strict threshold of twofold. In particular, NE cells had an extremely low fraction of cycling cells (0.127%), while B and plasma cells had a high fraction of cycling cells in epithelial-like SiNETs (32.25%) but not in neuronal-like SiNETs (0.178%).

## Within cell-type analysis

In each tumor, we examined clusters that were annotated as the same cell type but were not grouped together in the initial clustering analysis. For each such pair of clusters, we defined differential expression (with fold change greater than 3 and p-value lower than 0.05) and analyzed the expression of previously defined meta-programs associated with the corresponding cell type. FDR correction was applied to the p-values of individual genes to account for multiple comparisons.

## Comparing gene expression between SiNET subtypes across cell types

We performed a comparative gene expression analysis, examining epithelial-like and neuronal-like subtypes of SiNETs. The significance threshold for gene selection was established at an uncorrected p-value <0.05. Due to the limited sample size, we did not apply an FDR correction but added a second strict criterion of fold change >4. Additionally, we extended this analysis to bulk data, where each sample was assigned to the subtype for which the signature score was higher.

## Cell cycle correlations with cell type signatures in bulk SiNET samples

From GSE98894, we used the 81 SiNET samples. Counts were converted to TPM, and log2-transformed with an offset of 1. We then averaged the marker genes of epithelial, T cells, B cells, NE cells, and cell cycle to create scores of their expression. Few outlier samples with very low NE scores were excluded from further analysis. Finally, we calculated the Pearson correlation between the cell-type scores and the cell cycle score.

# Acknowledgements

This work was supported by grants from the Neuroendocrine Tumor Research Foundation (to IT and AT). IT is further supported by the Zuckerman STEM Leadership Program, the Mexican Friends New Generation, the Benoziyo Endowment Fund and E Harari. IT is the incumbent of the Dr. Celia Zwillenberg-Fridman and Dr. Lutz Zwillenberg Career Development Chair.

# Additional information

### Competing interests

Itay Tirosh: advisory board member of Immunitas Therapeutics, and a co-founder and advisory board member of Cellyrix Therapeutics. The other authors declare that no competing interests exist.

## Funding

| Funder | Grant reference number | Author |
| --- | --- | --- |
| Neuroendocrine Tumor Research Foundation | | Amit Tirosh Itay Tirosh |

The funders had no role in study design, data collection and interpretation, or the decision to submit the work for publication.

## Author contributions

Einav Somech, Software, Formal analysis, Methodology, Writing – original draft, Writing – review and editing; Debdatta Halder, Conceptualization, Data curation, Investigation, Methodology, Writing – original draft, Writing – review and editing; Avishay Spitzer, Conceptualization, Software, Formal analysis, Investigation, Methodology, Writing – original draft, Writing – review and editing; Chaya Barbolin, Michael Tyler, Formal analysis, Methodology; Reut Halperin, Writing – original draft, Writing – review and editing; Moshe Biton, Funding acquisition, Methodology; Amit Tirosh, Conceptualization, Resources, Supervision, Funding acquisition, Investigation, Writing – original draft, Project administration, Writing – review and editing; Itay Tirosh, Conceptualization, Formal analysis, Supervision, Investigation, Methodology, Writing – original draft, Project administration, Writing – review and editing

## Author ORCIDs

Einav Somech ⑩ https://orcid.org/0000-0002-3665-3105
Debdatta Halder ⑩ https://orcid.org/0009-0004-0780-7832
Avishay Spitzer ⑩ https://orcid.org/0000-0001-6551-9338
Reut Halperin ⑩ https://orcid.org/0000-0002-8137-667X
Amit Tirosh ⑩ https://orcid.org/0000-0003-3794-9634
Itay Tirosh ⑩ https://orcid.org/0000-0001-5477-2987

## Ethics

Human subjects: The study was approved by the Institutional Review Board of Sheba Medical Center (Ethical Approval Identifier: SMC-18-5674). Informed consent was obtained from each patient prior to any study-related procedures. Patients received a comprehensive explanation of the study's objectives and methods, were given ample time to consider their participation, and had the opportunity to ask questions and receive thorough answers before signing the consent form. Following the consent process, all samples and data were fully anonymized. Each participant was assigned a unique code, which was used throughout the study and was kept separate from any identifying information. As a result, the dataset is fully anonymized, and its use and publication do not require additional consent.

Reviewer #1 (Public review): https://doi.org/10.7554/eLife.101153.3.sa1
Reviewer #2 (Public review): https://doi.org/10.7554/eLife.101153.3.sa2
Reviewer #3 (Public review): https://doi.org/10.7554/eLife.101153.3.sa3
Author response https://doi.org/10.7554/eLife.101153.3.sa4

# Additional files

## Supplementary files

Supplementary file 1. Table of patient samples analyzed and their clinical characteristics.

Supplementary file 2. Table of cell type markers used to annotate clusters of cells.

Supplementary file 3. Tables listing the genes upregulated in neuroendocrine cells (compared to other cells in the same tumors), divided into those commonly found as upregulated across most SiNETs, those found only in a subset and those found in a specific tumor.

Supplementary file 4. Table per cell type, listing the differentially expressed genes of that cell type between the two small intestine neuroendocrine tumor (SiNET) subsets (Epithelial-like and Neuronal-like, abbreviated as EPI and NEU). Only genes with at least sixfold differential expression are listed. For each of those genes, log2-fold change, t-test p-value, and FDR-adjusted p-values are listed, and genes with unadjusted p-value below 0.05 were considered as significant and marked in red. An additional table lists the genes which were found as differentially expressed (between SiNET subsets) in all of the cell types.

Supplementary file 5. Table of gene signatures for malignant subpopulations identified in a sample of Large Cell Neuroendocrine Carcinoma (LCNEC) mixed with squamous cell carcinoma.
MDAR checklist

## Data availability

Single cell and single nuclei RNA-seq data is available through GEO with accession number GSE292163.

The following dataset was generated:

| Author(s) | Year | Dataset title | Dataset URL | Database and Identifier |
|---|---|---|---|---|
| Tirosh I, Somech E, Hadler D, Spitzer A, Brabolin C, Tyler M, Halperin R, Moshe B, Tirosh A | 2025 | ScRNA-seq of Small Intestine Neuroendocrine Tumors | https://www.ncbi.nlm.nih.gov/geo/query/acc.cgi?acc=GSE292163 | NCBI Gene Expression Omnibus, GSE292163 |

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
