## [Editor Report · eLife Assessment]

This is a **valuable** study that uses single-cell RNA sequencing to define tumor-intrinsic transcriptional programs that characterize distinct types of small intestine neuroendocrine tumors. The evidence supporting the claims of the authors is **solid**, but would benefit from a larger sample size. The work will be of interest to cancer biologists studying neuroendocrine tumors, as well as those studying tumor heterogeneity more broadly.

---

## [Referee Report · Reviewer #1 (Public review)]

Summary:

The authors have assembled a cohort of 10 SiNET, 1 SiAdeno, and 1 lung MiNEN samples to explore the biology of neuroendocrine neoplasms. They employ single-cell RNA sequencing to profile 5 samples (siAdeno, SiNETs 1-3, MiNEN) and single-nuclei RNA sequencing to profile seven frozen samples (SiNET 4-10).

They identify two subtypes of siNETs, characterized by either epithelial or neuronal NE cells, through a series of DE analyses. They also report findings of higher proliferation in non-malignant cell types across both subtypes. Additionally, they identify a potential progenitor cell population in a single lung MiNEN samples.

In the revised study, they have addressed my points and I have no further comments.

---

## [Referee Report · Reviewer #2 (Public review)]

Summary:

The research identifies two main SiNET subtypes (epithelial-like and neuronal-like) and reveals heterogeneity in non-neuroendocrine cells within the tumor microenvironment. The study validates findings using external datasets and explores unexpected proliferation patterns. While it contributes to understanding SiNET oncogenic processes, the limited sample size and depth of analysis present challenges to the robustness of the conclusions.

Strengths:

The studies effectively identified two subtypes of SiNET based on epithelial and neuronal markers. Key findings include the low proliferation rates of neuroendocrine (NE) cells and the role of the tumor microenvironment (TME), such as the impact of Macrophage Migration Inhibitory Factor (MIF).

Weaknesses:

However, the analysis faces challenges such as a small sample size and lack of clear biological interpretation in some analyses.

---

## [Referee Report · Reviewer #3 (Public review)]

This study profiles small intestine NETs and one mixed lung NET at single cell resolution and identifies two subtypes of neuroendocrine cells, as well as explores the proliferation patterns in malignant and nonmalignant cell types, identifying MIF as a potential factor that promotes proliferation of B and plasma cells in siNETs. Furthermore, they explore the single-cell landscape of a mixed LCNEC and squamous cell carcinoma, from which they identify a putative stem cell population with expression of features from both lineages.

Strengths:

This work showcases single-cell profiling of a rare tumor type, which is very informative for the field of NETs. The authors highlight very interesting observations, including the identification of the epithelial and neuronal subtype of siNETs, which they validated with an independent bulk RNA sequencing cohort. Furthermore, the observation of low cycling in malignant cells and high cycling in nonmalignant cells is an interesting one which may be applicable to other NETs.

Weaknesses:

• The authors do not connect their findings to clinical outcome. For example, is the epithelial or neuronal subtype enriched in tumors with worse or better prognosis or high grade vs. low grade siNETs or in patients who metastasize vs. who don't? As the authors show they can identify epithelial vs. neuronal subtypes in bulk RNA seq, perhaps they can take advantage of these other studies with larger sample sizes to investigate this. Additionally, the authors identify that the phenomenon of higher B/plasma cell proliferation is particular to epithelial siNETs and write that "The implications of high B/plasma cell turnover, and of other downstream effects of high MIF expression, are unclear, but raise the possibility that MIF-CD74 interaction may constitute a relevant target for the epithelial-like SiNET subtype." However, if this interaction contributes to survival in these patients, targeting this interaction may not be beneficial. Thus, it is important for the authors to try to connect their finding to clinical outcomes to enhance the translational relevance of this paper.

• The generalizability of this study would be enhanced if the authors analyzed other available single cell studies of NETs and found a similar phenomenon of high proliferating nonmalignant cell types. Although these studies are also very limited in sample size, seeing concordance in findings across independent cohorts and different experimental techniques would help to strengthen the findings. While the authors rationalize that these other studies are too distinct from their own due to enrichment for immune cells, this limitation should be noted but does not prevent such an analysis from being attempted.

• On page 3, the authors claim that "Technical effects (e.g. single cell analysis of fresh samples vs. single nuclei analysis of frozen samples) could also impact the capture of distinct cell types, although we did not observe a clear pattern of such bias." Can the authors show that cell type frequencies are not significantly different between the samples profiled with these two methods?

• Why did siNET3 and siNET9 have much lower recovery of neuroendocrine cells compared to other samples? It would be interesting to see how similar or different the transcriptional profiles are of the samples that were obtained from the same patient, considering that multifocal siNETs are found to derive from distinct clones, although this analysis is understandably not possible in this case due to the lack of neuroendocrine cells in one of two samples from the same patient.

• It should be more clearly stated in the text that these samples were previously treated with somatostatin analogues, as this impacts the interpretation of the findings.

• The identification of a potential progenitor subtype in the miNEN is very intriguing, albeit a case study and represents a distinct cancer from the lowly proliferating siNETs. While the authors mention this in the text, the case study feels rather tangential to the other parts of the paper.

• How the authors compared the DE genes to known signatures for the fibroblast and endothelial cells should be clarified and discussed in the Methods section.

---

## [Author Response]

The following is the authors’ response to the original reviews

**Public Reviews:**

**Reviewer #1 (Public review):**
Summary:The authors have assembled a cohort of 10 SiNET, 1 SiAdeno, and 1 lung MiNEN samples to explore the biology of neuroendocrine neoplasms. They employ single-cell RNA sequencing to profile 5 samples (siAdeno, SiNETs 1-3, MiNEN) and single-nuclei RNA sequencing to profile seven frozen samples (SiNET 4-10).

They identify two subtypes of siNETs, characterized by either epithelial or neuronal NE cells, through a series of DE analyses. They also report findings of higher proliferation in non-malignant cell types across both subtypes. Additionally, they identify a potential progenitor cell population in a single-lung MiNEN sample.

Strengths:Overall, this study adds interesting insights into this set of rare cancers that could be very informative for the cancer research community. The team probes an understudied cancer type and provides thoughtful investigations and observations that may have translational relevance.Weaknesses:The study could be improved by clarifying some of the technical approaches and aspects as currently presented, toward enhancing the support of the conclusions:(1) Methods: As currently presented, it is possible that the separation of samples by program may be impacted by tissue source (fresh vs. frozen) and/or the associated sequencing modality (single cell vs. single nuclei). For instance, two (SiNET1 and SiNET2) of the three fresh tissues are categorized into the same subtype, while the third (SiNET9) has very few neuroendocrine cells. Additionally, samples from patient 1 (SiNET1 and SiNET6) are separated into different subtypes based on fresh and frozen tissue. The current text alludes to investigations (i.e.: "Technical effects (e.g., fresh vs. frozen samples) could also impact the capture of distinct cell types, although we did not observe a clear pattern of such bias."), but the study would be strengthened with more detail.

We thank the reviewer for the thoughtful and constructive review. Due to the difficulty in obtaining enough SiNET samples, we used two platforms to generate data - single cell analysis of fresh samples, and single nuclei analysis of frozen samples. We opted to combine both sample types in our analysis while being fully aware of the potential for batch effects. We therefore agree that this is a limitation of our work, and that differences between samples should be interpreted with caution.

Nevertheless, we argue that the two SiNET subtypes that we have identified are very unlikely to be due to such batch effect. First, the epithelial SiNET subtype was not only detected in two fresh samples but also in one frozen sample (albeit with relatively few cells, as the reviewer correctly noted). Second, and more importantly, the epithelial SiNET subtype was also identified in analysis of an external and much larger cohort of bulk RNA-seq SiNET samples that does not share the issue of two platforms (as seen in Fig. 2f). Moreover, the proportion of samples assigned to the two subtypes is similar between our data and the external data. We therefore argue that the identification of two SiNET subtypes cannot be explained by the use of two data platforms. However, we agree that the results should be further investigated and validated by future studies.

The reviewer also commented that two samples from the same patient which were profiled by different platforms (SiNET1 and SiNET6) were separated into different subtypes. We would like to clarify that this is not the case, since SiNET6 was not included in the subtype analysis due to too few detected Neuroendocrine cells, and was not assigned to any subtype, as noted in the text and as can be seen by its exclusion from Figure 2 where subtypes are defined. We apologize that our manuscript may have given the wrong impression about SiNET6 classification (it was labeled in Fig. 4a in a misleading manner). In the revised manuscript, we corrected the labeling in Fig. 4a and clarified that SiNET6 is not assigned to any subtype. We also further acknowledge the limitation of the two platforms and the arguments in favor of the existence of two SiNET subtypes.

(Additional specific recommendations for the authors are provided below)(2) Results:Heterogeneity in the SiNET tumor microenvironment: It is unclear if the current analysis of intratumor heterogeneity distinguishes the subtypes. It may be informative if patterns of tumor microenvironment (TME) heterogeneity were identified between samples of the same subtype. The team could also evaluate this in an extension cohort of published SiNET tumors (i.e. revisiting additional analyses using the SiNET bulk RNAseq from Alvarez et al 2018, a subset of single-cell data from Hoffman et al 2023, or additional bulk RNAseq validation cohorts for this cancer type if they exist [if they do not, then this could be mentioned as a need in Discussion])

We agree that analysis of an independent cohort will assist in defining the association between TME and the SiNET subtype. However, the sample size required for that is significantly larger than the data available. In the revised manuscript we note that as a direction for future studies.

(3) Proliferation of NE and immune cells in SiNETs: The observed proliferation of NE and immune cells in SiNETs may also be influenced by technical factors (including those noted above). For instance, prior studies have shown that scRNA-seq tends to capture a higher proportion of immune cells compared to snRNA-seq, which should be considered in the interpretation of these results. Could the team clarify this element?

We agree that different platforms could affect the observed proportions of immune cells, and more generally the proportions of specific cell types. However, the low proliferation of Neuroendocrine cells and the higher proliferation of immune cells (especially B cells, but also T cells and macrophages) is consistently observed in both platforms, as shown in Fig. 4a, and therefore appears to be reliable despite the limitations of our work. We clarify this consistency in the revised manuscript.

(4) Putative progenitors in mixed tumors: As written, the identification of putative progenitors in a single lung MiNEN sample feels somewhat disconnected from the rest of the study. These findings are interesting - are similar progenitor cell populations identified in SiNET samples? Recognizing that ideally additional validation is needed to confidently label and characterize these cells beyond gene expression data in this rare tumor, this limitation could be addressed in a revised Discussion.

We do not find evidence for similar progenitors in the SiNET samples, but they also do not contain two co-existing lineages of cancer cells within the same tumor, so this is harder to define. We agree about the need for additional validation for this specific finding and have noted that in the revised Discussion.

**Reviewer #2 (Public review):**
Summary:The research identifies two main SiNET subtypes (epithelial-like and neuronal-like) and reveals heterogeneity in non-neuroendocrine cells within the tumor microenvironment. The study validates findings using external datasets and explores unexpected proliferation patterns. While it contributes to understanding SiNET oncogenic processes, the limited sample size and depth of analysis present challenges to the robustness of the conclusions.Strengths:The studies effectively identified two subtypes of SiNET based on epithelial and neuronal markers. Key findings include the low proliferation rates of neuroendocrine (NE) cells and the role of the tumor microenvironment (TME), such as the impact of Macrophage Migration Inhibitory Factor (MIF).Weaknesses:However, the analysis faces challenges such as a small sample size, lack of clear biological interpretation in some analyses, and concerns about batch effects and statistical significance.
**Reviewer #3 (Public review):**
Summary:In this study, the authors set out to profile small intestine neuroendocrine tumors (siNETs) using single-cell/nucleus RNA sequencing, an established method to characterize the diversity of cell types and states in a tumor. Leveraging this dataset, they identified distinct malignant subtypes (epithelial-like versus neuronal-like) and characterized the proliferative index of malignant neuroendocrine cells versus non-malignant microenvironment cells. They found that malignant neuroendocrine cells were far less proliferative than some of their non-malignant counterparts (e.g., B cells, plasma cells, epithelial cells) and there was a strong subtype association such that epithelial-like siNETs were linked to high B/plasma cell proliferation, potentially mediated by MIF signaling, whereas neuronal-like siNETs were correlated with low B/plasma cell proliferation. The authors also examined a single case of a mixed lung tumor (neuroendocrine and squamous) and found evidence of intermediate/mixed and stem-like progenitor states that suggest the two differentiated tumor types may arise from the same progenitor.Strengths:The strengths of the paper include the unique dataset, which is the largest to date for siNETs, and the potentially clinically relevant hypotheses generated by their analysis of the data.Weaknesses:The weaknesses of the paper include the relatively small number of independent patients (n = 8 for siNETs), lack of direct comparison to other published single-cell NET datasets, mixing of two distinct methods (single-cell and single-nucleus RNA-seq), lack of direct cell-cell interaction analyses and spatially-resolved data, and lack of in vitro or in vivo functional validation of their findings.The analytical methods applied in this study appear to be appropriate, but the methods used are fairly standard to the field of single-cell omics without significant methodological innovation. As the authors bring forth in the Discussion, the results of the study do raise several compelling questions related to the possibility of distinct biology underlying the epithelial-like and neuronal-like subtypes, the origin of mixed tumors, drivers of proliferation, and microenvironmental heterogeneity. However, this study was not able to further explore these questions through spatially-resolved data or functional experiments.
**Recommendations for the authors:**

**Reviewer #1 (Recommendations for the authors):**
(1) Methods:a) Could the team clarify the discrepancy in subtype assignment between two samples from the same patient? i.e. are these samples from the same tumor? If so, what does the team think is the explanation for the difference in subtype assignment?

As noted above in response to the public review of reviewer #1, SiNET6 was in fact not assigned to any subtype (due to insufficient NE cells) and hence there was no discrepancy. We apologize for the misleading labeling of SiNET6 in the previous version and have corrected this In the revised version of Figure 4.

b) What is the rationale for scoring tumor-derived programs on samples with no tumor cells? For instance, SiNET3 does not contain NE cells, and SiNET9 has a very low fraction of NE cells. Please clarify how the scoring was performed on these samples, as the program assignments may be driven by other cell types in samples with little to no NE cells.

Scoring for tumor-derived programs was done only for the NE cells. Accordingly, SiNET3 was not scored or assigned to any of the programs. SINET9 was included in this analysis - although it had a relatively small fraction of NE cells, the absolute number of profiled cells was particularly high in this sample and therefore the number of NE cells was 130, higher than our cutoff of 100 cells.

c) Given the heterogeneity of cell types within each sample, would there be a way to provide a refined sense of confidence for certain cell type annotations? This would be helpful given the heterogeneity in marker gene expression and the absence of gold-standard markers for fibroblasts and endothelial cells in this cancer type. Additionally, there seems to be an unusually large proportion of NK and T cells - was there selection for this (given that these tumors are largely not immune infiltrated)?

Author Response: Except for the Neuroendocrine cells, there are six TME cell types that we consistently find in multiple SiNET samples: macrophages, T cells, B/plasma cells, fibroblasts, endothelial and epithelial cells. Each of these cell types are identified as discrete clusters in analysis of the respective tumors (as shown in Fig. 1a,b and Fig. S1), and these are exactly the six most common non-malignant cell types that we and others found in single cell analysis across various other tumor types (e.g. see Gavish et al. 2023, ref. #15). The signatures used to annotate these cell types are shown in Supplementary file 2, and they primarily consist of classical markers that are traditionally used to define those cell types. We therefore believe that the annotation of these typical tumor-associated cell types is robust and does not include major uncertainties. In addition to these five common cell types, there are three cell types that we find only in 1-2 of the samples – epithelial cells, plasma cells and NK cells. Again, we believe that their annotation is robust, and these cell types are primarily not used for further analysis.

There was no selection for any specific cell types in this study. Nevertheless, single cell (or single nuclei) analysis may lead to biases towards specific cell types, that we cannot evaluate directly from the data. NK cells were detected only in one tumor. T cells were detected in eight of the ten samples; but in four of those samples the frequency of T cells was lower than 5% and only in one sample the frequency was above 20%. Therefore, while we cannot exclude a technical bias towards high frequency of T/NK cells, we do not consider these frequencies as high enough to suggest this specific type of bias. In the revised manuscript, we clarify that the commonly observed cell types in SiNETs are the same as those commonly observed in other tumors and we acknowledge the possibility of a technical bias in cell type capture.

d) Evaluating the expression of one gene at a time may not effectively demonstrate subtype-specific patterns, particularly when comparing NE cells from one tumor to non-NE cells from another, which may not be an appropriate approach for identifying differentially expressed genes. DE analysis coupled with concordance analysis, for example, could strengthen the results.

We apologize, but we do not fully understand this comment. We note that the initial normalization by non-NE cells was done in order to decrease batch effects when combining the data of the two platforms. We also note that the two subtypes were identified by two distinct approaches, as shown in Fig. 2c and in Fig. 2f.

(2) Results:See the above public review.(3) Minor Comments:a) Results: Single cell and single nuclei RNA-seq profiling of SiNETsThe results say ten primary tumor samples from eight patients. Later in the paragraph it says, "After initial quality controls, we retained 29,198 cells from the ten patients." Please clarify to either ten samples or eight patients.

Indeed these are ten samples rather than ten patients. We corrected that in the revised version and thank the reviewer for noticing our error.

b) Methods:- Please specify which computational tools were used to perform quality control, signature scoring, etc.

The approaches for quality control, scoring etc. are described in the methods. We implemented these approaches with R code and did not use other computational tools.

- Minor point but be consistent with naming convention (ie, siAdeno vs SiAdeno) throughout the paper. For example, under "Sample Normalization, Filtering and annotations" change "siAdeno" to "SiAdeno."

Thank you for noting this, we corrected that.

- Add processing and analysis of MiNEN sample to the methods section. It is not mentioned in the methods at all.

As noted in the revised manuscript, the MiNEN sample was analyzed in the same way as the SiNET fresh samples.

c) Supplementary Figures:Figure S1: Change (A-H) to (A-I) to account for all panels in the figure.Figure S4: Add (C) after "the siAdeno sample" in the legend.

Thank you for noting this, we corrected that.

(4) Font size is quite small in the main figures.

We enlarged the font in selected figure panels.

**Reviewer #2 (Recommendations for the authors):**
(1) The small number of samples used in some analyses affects the robustness of the findings. Increasing the sample size or including more validation data could improve the statistical reliability and make the results more convincing. The authors should consider expanding the cohort size or integrating additional external datasets to increase statistical power.

We agree with the reviewer that adding more samples would improve the reliability of the results. However, the external data that we found was not comparable enough to enable integration with our data, and we are unable to profile additional SiNET samples in our lab. We hope that future studies would support our results and extend them further.

(2) The biological significance of differentially expressed genes needs more depth, limiting the insights into SiNET biology. The authors should perform a comprehensive pathway enrichment analysis and integrate findings with existing literature. Tools like Gene Set Enrichment Analysis (GSEA) or Overrepresentation Analysis (ORA) could provide a more holistic view of altered biological processes.

We thank the reviewer for this suggestion. We did examine the functional enrichment of differentially expressed genes and did not find additional enrichments that we felt were important to highlight beyond what we described. We report the genes in supplementary files, enabling other researchers to examine these lists further.

(3) The unexpected finding of higher proliferation in non-malignant cells requires further investigation and plausible biological explanation. The authors should perform additional analyses to explore potential mechanisms, such as investigating cell cycle regulators or performing in vitro validation experiments. The authors should consider single-cell trajectory analysis to explore these highly proliferative non-malignant cells' potential differentiation or activation states.

We agree that our results are descriptive and that we do not fully explain the mechanism for the high level of non-malignant cell proliferation. We did attempt to perform follow up computational analysis. These analyses raised the hypothesis that high levels of MIF are causing the proliferation of immune cells. Additional analyses that we performed were not sufficient to conclusively identify a mechanism, and we felt that they were not informative enough to be included in the manuscript. Further *in vitro* (or *in vivo*) studies are beyond the scope of the current work.

(3) More details are required on methods used for p-value adjustment, and criteria for statistical significance should be clearly defined. Additionally, integrating scRNA-seq and snRNA-seq data needs a more thorough explanation, including batch effect mitigation and more explicit cell clustering representation. The authors should clearly describe p-value adjustments (e.g., FDR) and batch correction methods (e.g., Harmony, FastMNN integration) and include additional figures showing corrected UMAP plots or heatmaps post-batch correction to enhance the confidence in results.

We now clarify in the Methods our use of FDR for p-value adjustments. As for batch correction, we have avoided the use of integration methods as we believe that they tend to distort the data and decrease tumor-specific signals. Instead, we primarily analyzed one tumor at a time and never directly compared cell profiles across distinct tumors but only compared the differences between subpopulations; specifically, we normalized the expression of NE cells by subtracting the expression of reference non-NE cells from the same tumor as a method to decrease batch effects. We now clarify this point in the Methods section.

(4) The lack of analysis of interactions between different cell types limits understanding of tumor microenvironment dynamics. The authors should employ cell-cell interaction analysis tools (e.g., CellPhoneDB, NicheNet) to explore potential communication networks within the tumor microenvironment. This could provide valuable insights into how different cell types influence tumor progression and maintenance.

We thank the reviewer for this suggestion. We have tried to use such methods but found the results difficult to interpret since these approaches generated very long lists of potential cell-cell interactions that are largely not unique to the SiNET context and their relevance remains unclear without follow up experiments, which are beyond the scope of this work. We therefore focused only on ligand/receptors that came up robustly through specific analyses such as the differences between SiNET subtypes. In particular, MIF is highly expressed in the epithelial subtype, and remarkably, MIF upregulation is shared across multiple cell types. Thus, the cell-cell interactions that are suggested by the SiNET data as somewhat unique to this context are those involving MIF and its receptor (CD74 on immune cell types), while other interactions detected by the proposed methods primarily reflect the generic ligand/receptors expressed by corresponding TME cell types.

**Reviewer #3 (Recommendations for the authors):**
(1) For a relatively small dataset, the mixing of single-cell versus single-nucleus RNA-seq should be discussed more. It would be nice to have 1-2 tumors that are analyzed by both methods to compare and increase our understanding of how these different approaches may affect the results. This could be accomplished by splitting a fresh tumor into two parts, processing it fresh for single-cell RNA-seq, and freezing the other part for single-nucleus RNA-seq.

We agree with the reviewer that the different techniques may bias our results and we refer to this limitation in the Results and Discussion sections. However, it is important to note that we do not directly integrate the primary data across these modalities, but rather analyze each tumor separately and only combine the results across tumors. For example, we first compare the NE cells from each tumor to control non-NE cells from the same tumor and then only compare the sets of NE-specific genes across tumors. Moreover, the subtypes that we detect cannot be explained by these modalities, as the first subtype contains samples from both methods and these subtypes are further demonstrated in external bulk data. Similarly, the results regarding low proliferation of NE cells and high proliferation of B/plasma cells are observed across both modalities. We therefore argue that while the combination of methods is a limitation of this work it does not account for the main results.

(2) The authors state that they defined the siNET transcriptomic signature by comparing their siNET single-cell/nucleus data to other NETs profiled by bulk RNA-seq. Some of the genes in the signature, such as CHGA, are widely used as markers for NETs (and not specific for siNET). The authors should address this in more detail.

To define the SiNET transcriptomic signature we first analyzed each tumor separately and compared the expression of Neuroendocrine (NE) cells to that of non-NE cells to detect NE-specific genes. Next, we compared the lists of NE-specific genes across the 8 SiNET patients and found a subset of 26 genes which were shared across most of the analyzed SiNET samples (Fig. 2a). Thus, the signature was defined only from analysis of SiNETs and not based on comparison to other types of NETs and hence it is expected that the signature could contain both SiNET-specific genes and more generic NET genes such as CHGA.

Only after defining this signature, we went on to compare it between SiNETs and other types of NETs (pancreatic and rectal) based on external bulk RNA-seq data. In this comparison, we observed that the signature was clearly higher in SiNETs than in the other NETs (Fig. 2b). This result supports the accuracy of the signature and further suggests that it contains a fraction of SiNET-specific genes and not only generic NET genes such as CHGA. Thus, we would expect this signature to perform well also for distinguishing between SiNET and types of NETs, but it does contain a subset of genes that would be high in the other NETs. Finally, we note that even though CHGA is a generic NET marker, the bulk RNA-seq data would suggest that, at least at the mRNA level, this gene is still higher expressed in SiNETs than in other NETs. To avoid confusion regarding the definition and specificity of the SiNET transcriptomic signature we have extended the description of this section in the revised manuscript.(3) The authors only compare their data to bulk transcriptomic data on NETs. While in some instances this makes sense given the bulk dataset has >80 tumors, they should at least cite and do some comparison to other published single-cell RNA-seq datasets of NETs (e.g., PMID: 37756410, 34671197). The former study listed has 3 siNETs, 4 pNETs, and 1 gNET. Do the epithelial-like and neuronal-like signatures show up in this dataset too?

We examined these studies but concluded that their data was inadequate to identify the two SiNET subtypes. The latter study was of pNETs, while the former study had 3 SiNET samples but only from 2 patients, and furthermore it was enriching for immune cells with only very low amounts of NE cells. Therefore, we now cite this work in the discussion but cannot use it to extend the results from our work.

(4) How did the authors statistically handle patients with more than one tumor sample (true for n = 2)? These tumor samples would not be truly independent.

In both cases where we had two distinct samples of the same patient, only one sample had sufficient NE cells to be included in NE-related analysis and therefore the other samples (SiNET3 and SiNET6) were excluded from all analysis of NE differential expression and subtypes. These samples were only included in the initial analysis (Fig. 1) and in TME-related analysis (Fig. 3-4) in which there was no statistical analysis of differences between patients and hence no problem with the inclusion of 2 samples for the same patient. We clarified this issue in the revised version.

(5) The association between siNET subtype and B/plasma cell proliferation is very interesting, as is the hypothesis regarding MIF signaling. It would be illuminating for the authors to perform cell-cell interaction analyses with methods such as CellChat in this context rather than just relying on DE. Spatial mapping would be helpful too and while this may be outside the scope of this study, it should at least be expounded upon in the Discussion section.

Indeed, spatial transcriptomic analysis would add interesting insight to our data and to SiNET biology. Unfortunately, this is not within the scope of the current project but we note this interesting possibility in the Discussion. Regarding additional methods for cell-cell interactions, we have performed such analysis but found it not informative as it highlighted a large number of interactions that are not unique SiNETs and are difficult to interpret, and therefore we do not include this in the revised version.

(6) The authors note that in the mixed lung tumor, the NE component was more proliferative than that observed with siNETs. How does the proliferation compare to pNETs, gNETs, in other published studies? How about assessing the clonality of the SCC and LNET malignant cells with various genomic or combined genomic/transcriptomic methods?

The percentage of proliferating NE cells in the mixed lung tumor was higher than 60%. This is extremely high, approximately four-fold higher than the average that we found in a pan-cancer analysis and higher than the average of any of the >20 cancer types that we analyzed (Gavish et al. 2023, ref. #15). This remarkably high proliferation serves as a control for the low proliferation that we found in SiNET NE cells.

(7) In the Discussion on page 13, the authors write "Second, proliferation of NE cells may be inhibited by prior treatments with somatostatin analogues." How many patients were treated in this manner? This information should be made more explicit in the manuscript.

Details on pretreatment with somatostatin analogues are provided in Supplementary file 1. All patients were pre-pretreated with somatostatin analogues, with the possible exception of one patient (P8, SiNET10) for which we could not confidently obtain this information.

(8) On page 5, "bone-fide" is misspelled.(9) On page 8, "exact identify" is misspelled.

We thank the reviewer and have corrected the typos.